# Look what you make my tissues do: The role of metalloproteinases and their inhibitors in *Bothrops* snakebites

Juliana Costa Ferreira Neves[1,2], Fábio Magalhães-Gama[3,4],
Hiochelson Najibe Santos Ibiapina[1,2], Kamille Beltrão Seixas[3,5], Êndila Souza Barbosa[1,2],
Adriana Malheiro[3,5,6], Jacqueline Almeida Gonçalves Sachett[1,2,7],
Ana Carolina Campi-Azevedo[4], Olindo Assis Martins-Filho[4,6], Andréa Teixeira-Carvalho[4,6],
Marco Aurélio Sartim[1,2,5], Wuelton Monteiro[1,2,8], Allyson Guimarães Costa [1,3,5,6]*

**1** Programa de Pós-Graduação em Medicina Tropical, Universidade do Estado do Amazonas (UEA), Manaus, Brazil, **2** Diretoria de Ensino e Pesquisa, Fundação de Medicina Tropical Heitor Vieira Dourado (FMT-HVD), Manaus, Brazil, **3** Diretoria de Ensino e Pesquisa, Fundação Hospitalar de Hematologia e Hemoterapia do Amazonas (HEMOAM), Manaus, Brazil, **4** Grupo Integrado de Pesquisas em Biomarcadores, Instituto René Rachou, Fundação Oswaldo Cruz (FIOCRUZ-Minas), Belo Horizonte, Brazil, **5** Programa de Pós-Graduação em Imunologia Básica e Aplicada, Instituto de Ciências Biológicas (ICB), Universidade Federal do Amazonas (UFAM), Manaus, Brazil, **6** Programa de Pós-Graduação em Ciências Aplicadas à Hematologia, Universidade do Estado do Amazonas (UEA), Manaus, Brazil, **7** Programa de Pós-Graduação em Enfermagem em Saúde Pública (ProEnsp), Universidade do Estado do Amazonas (UEA), Manaus, Brazil, **8** Duke Global Health Institute, Duke University, Durham, United States of America

* allyson.gui.costa@gmail.com

## Abstract

### Background

*Bothrops* envenomation induces extensive local tissue destruction and a robust inflammatory response, largely driven by the host's endogenous molecular pathways. Among these, Matrix Metalloproteinases (MMPs), zinc-dependent endopeptidases responsible for extracellular matrix (ECM) degradation, play a central role. The clinical severity of envenomation is therefore strongly influenced by the balance between MMPs and their specific inhibitors, the TIMPs. This study investigated the contribution of MMP-1, MMP-2, MMP-7, MMP-9, and MMP-10 and TIMP-1, TIMP-2, TIMP-3, and TIMP-4 to the inflammatory response following *Bothrops* snakebites.

### Methods and findings

In this study, we prospectively enrolled 30 patients, classified them as Mild or Severe, and quantified circulating MMPs and TIMPs concentrations before and after antivenom administration using a multiplex Luminex platform. Early inflammatory markers and initial MMPs activation (MMP-2, MMP-7, MMP-9, MMP-10) did not differ significantly between groups. However, post-antivenom molecular trajectories diverged sharply. Mild cases exhibited effective enzymatic regulation, restoring the

**Data availability statement:** Due to ethical restrictions related to patient privacy and the approval granted by the Research Ethics Committee of Fundação de Medicina Tropical Dr. Heitor Vieira Dourado (protocol no. 492.892), individual-level data cannot be made publicly available. De-identified data underlying the findings of this study are available upon reasonable request to the Ethics Committee of Fundação de Medicina Tropical Dr. Heitor Vieira Dourado (FMT-HVD) by email (cep@fmt.am.gov.br) for researchers who meet the criteria for access to confidential data.

**Funding:** Financial support was provided in the form of grants from Coordenação de Aperfeiçoamento de Pessoal de Nível Superior (CAPES) (Funding Code 001 to JCFN, PROAP Program #1247/2022 to WM, and PDPG CONSOLIDACAO-3-4 Program #88887.707248/2022-00 to AGC), Fundação de Amparo à Pesquisa do Estado do Amazonas (FAPEAM) (PDPG/CAPES/FAPEAM Program #038/2022 to WM, and POSGRAD Program #002/2025 and #015/2026 to JCFN), Fundacão de Amparo à Pesquisa do Estado de Minas Gerais (FAPEMIG) (grant #APQ-00432-20 and APQ-01499-21 to OAM-F), and Conselho Nacional de Desenvolvimento Científico e Tecnologico (CNPq). JAGS, OAM-F, AT-C, WMM, and AGC are grateful to CNPq for their research fellowship (Produtividade Program). OAM-F and AT-C participated in the fellowship program supported by the FAPEAM (PV-PD CT&I/FAPEAM Program #002/2025). The funders had no role in study design, data collection and analysis, decision to publish, or preparation of the manuscript.

**Competing interests:** The authors have declared that no competing interests exist.

MMPs/TIMPs profile toward a state that favored ECM turnover and tissue repair. In contrast, Severe cases showed persistent dysregulation, with a sustained imbalance that hindered ECM reorganization and perpetuated damaging inflammatory pathways.

## Conclusion

These findings suggest that the regulation of MMP/TIMP balance following antivenom therapy may be associated with clinical evolution. Further studies are required to determine whether these molecular patterns can be validated as prognostic markers or therapeutic targets.

## Author summary

Snakebite envenomation caused by *Bothrops* snakes is a major public health problem in the Amazon and frequently leads to severe local tissue damage. Much of this injury is driven not only by the venom itself but also by the body's own inflammatory response. In this study, we examined a group of molecules called matrix metalloproteinases (MMPs) and their natural inhibitors (TIMPs), which help regulate how tissues are broken down and repaired. We followed 30 patients with mild or severe envenomation and measured these molecules before and after antivenom treatment. We found that patients start with similar levels of MMPs, but their responses after treatment differ greatly. Individuals with mild envenomation were able to re-establish a healthy balance between MMPs and TIMPs, supporting tissue repair. In contrast, severe cases showed persistent imbalance, which may worsen inflammation and prevent proper healing. Our findings suggest that monitoring these molecules may help identify patients at higher risk of complications and could guide new therapeutic approaches to reduce long-term damage caused by snakebites.

## Introduction

Snakebites remain a neglected public health problem and are responsible for substantial morbidity and mortality, particularly in tropical and subtropical regions of the world [1,2]. In Brazil, snakes of the genus *Bothrops* are widely distributed, comprising approximately 29 species that inhabit diverse environments ranging from savannas and agricultural areas to tropical forests and even peri-urban and urban settings [3,4]. Within the Brazilian Amazon, Bothrops atrox is the predominant species and accounts for nearly 90% of all reported envenomation cases [5–7].

*Bothrops* venoms consist of a complex mixture of bioactive toxins, primarily snake venom metalloproteinases (SVMPs), serine proteases (SPs), and phospholipases $A_2$ ($PLA_2$) [3,8,9]. These components are known to contribute to the characteristic clinical and pathophysiological features of envenomation, including acute pain, edema,

extensive local tissue damage, and potentially severe systemic disturbances [10]. Experimental studies indicate that specific venom fractions can elicit pronounced local alterations, largely mediated by SVMP activity. In addition to direct tissue injury, these toxins have been shown to trigger host inflammatory responses that may involve the activation of endogenous metalloproteinases, potentially amplifying local damage [11].

Endogenous molecular responses are therefore important contributors to the progression and severity of Bothrops envenomation. Among these responses, matrix metalloproteinases (MMPs) have attracted increasing attention. MMPs are a family of zinc-dependent endopeptidases capable of degrading major extracellular matrix (ECM) components, including collagen, elastin, proteoglycans, and fibronectin, and of processing a variety of bioactive molecules such as cytokines, chemokines, and cell-surface receptors [12]. Through these functions, MMPs participate in cellular migration, angiogenesis, vascular permeability, and several aspects of the inflammatory response, placing them at the interface between venom-induced injury and host-mediated tissue remodeling.

The activity of MMPs is tightly regulated by tissue inhibitors of metalloproteinases (TIMPs), which play a critical role in maintaining the balance between ECM synthesis and degradation. Disruption of the MMP–TIMP equilibrium, whether through excessive MMP activation or insufficient TIMP availability, has been associated with accelerated ECM breakdown, loss of vascular integrity, enhanced leukocyte infiltration, and amplification of pro-inflammatory mediators. These processes may contribute to extensive local necrosis and, in some cases, to impaired tissue repair or chronic fibrosis [12–14].

Given the proposed role of endogenous metalloproteinases in modulating venom-induced tissue injury, elucidating the interplay between MMPs and TIMPs in Bothrops envenomation is important for advancing our understanding of the molecular pathways associated with inflammation and tissue damage. Characterizing how this balance evolves during the clinical course of envenomation may provide insights into molecular pathways potentially associated with disease severity.

In this context, the present study prospectively examines circulating profiles of selected MMPs and TIMPs in patients with suspected Bothrops envenomation, stratified according to clinical severity. By evaluating their temporal dynamics before and after antivenom administration, this study aims to provide exploratory data on host metalloproteinase regulation during envenomation and its possible association with clinical outcomes.

## Materials and methods

### Ethical approval and consent to participate

This study was approved by the Research Ethics Committee of the Fundação de Medicina Tropical Dr. Heitor Vieira Dourado (FMT-HVD) (protocol no. 492.892). All participants received information about the objectives and procedures of the study and provided written informed consent prior to enrollment. The research procedures complied with the principles outlined in the Declaration of Helsinki and with Resolution 466/12 of the Brazilian National Health Council for research involving human subjects. In addition, the study was registered in the Brazilian Registry of Clinical Trials (ReBEC) (UTN code: U1111-1169–1005).

### Study design

To investigate these mechanisms in a real-world clinical context, we conducted an observational, longitudinal, and prospective study involving patients who experienced snakebites caused by *Bothrops* species and subsequently sought medical care at the Fundação de Medicina Tropical Dr. Heitor Vieira Dourado (FMT-HVD). To enable comparisons between envenomated individuals and those without inflammation related to snakebite, we also included a healthy donor (HD) control group, recruited in partnership with the Fundação Hospitalar de Hematologia e Hemoterapia do Amazonas (HEMOAM).

### Data collection and blood sampling

The study population consisted of 30 individuals who experienced suspected Bothrops spp. envenomation and sought medical care at FMT-HVD. Case identification was based on clinical and epidemiological criteria evaluated by the

attending medical team, according to Brazilian Ministry of Health guidelines. These criteria included local and systemic manifestations characteristic of Bothrops envenomation, which differ from those observed in other types of snakebite accidents prevalent in the Amazon region.

Patients were stratified into two clinical subgroups according to established criteria: Mild, characterized by limited local manifestations (edema and pain) without necrosis and with minor systemic signs (e.g., mild coagulopathy or bleeding, absence of shock); and Severe, defined by pronounced local manifestations (intense edema and pain) accompanied by abscess, blister formation, and/or necrosis, along with severe systemic alterations such as major bleeding, hypotension, shock, or acute kidney injury [15].

All patients were classified and managed according to Brazilian Ministry of Health guidelines and received bothropic antivenom (SAB) produced by Instituto Butantan [16]. Antivenom administration ranged from 2 to 4 vials for mild cases and up to 12 vials for severe cases, depending on clinical presentation. To establish a baseline for comparison, the study also included 20 healthy blood donors (HD) of both sexes, with no history of snakebite. Exclusion criteria comprised pregnancy, individuals under 18 years of age, participants from Indigenous communities, and those with self-reported inflammatory or immune-mediated conditions, including autoimmune diseases, immunodeficiencies, or coagulopathies.

Patients were monitored at three distinct time points: before antivenom administration (T0) and 24 and 48 hours after treatment (T1 and T2, respectively). The T0 time point corresponded to blood collection performed immediately after hospital admission and prior to antivenom administration, as part of the initial clinical evaluation. This sampling therefore reflects the early host response following envenomation, before the initiation of antivenom therapy.

At each time point, a 4 mL aliquot of peripheral blood was collected by venipuncture into EDTA tubes. Following collection, samples were kept at room temperature (20–25°C) and processed within 4 hours. Plasma was obtained after centrifugation at 900×g for 15 minutes, and the resulting supernatant was transferred to cryovials and stored at -80 °C until analysis of MMPs and TIMPs. Data collection for the HD group was performed at a single time point. Sociodemographic and epidemiological information was obtained through standardized questionnaires, and additional clinical data were retrieved from electronic medical records (iDoctor) at FMT-HVD. The overall study design and sample collection workflow are depicted in Fig 1.

## Quantification of soluble immunological mediators

For the quantification of circulating metalloproteinases and their inhibitors, plasma aliquots were thawed at 37°C and centrifuged at 24,000×g to remove lipid residues and particulate debris. The resulting supernatant was then passed through a 0.22 μm syringe filter to ensure sample clarity and minimize potential assay interference. Concentrations of metalloproteinases (MMP-1, MMP-2, MMP-7, MMP-9, MMP-10) and their endogenous inhibitors (TIMP-1, TIMP-2, TIMP-3, TIMP-4) were measured using the MILLIPLEX Human MMP Magnetic Bead Panel 2 (Cat. HMMP2MAG-55K) and MILLIPLEX Human TIMP Magnetic Bead Panel 2 (Cat. HTMP2MAG-54K), respectively. Sample acquisition was performed on a Luminex 200 system at the *Instituto René Rachou/ FIOCRUZ-Minas*. Data analysis of Median Fluorescent Intensity (MFI) values and corresponding concentrations (pg/mL) was conducted using the MANAGER software, BIORAD, employing either a five-parameter logistic (5-PL) or spline curve-fitting model for analyte quantification. Each patient sample represented an independent biological replicate. All measurements were performed in duplicate for each plasma sample, and duplicate measurements were considered technical replicates. The mean of duplicate values was used for subsequent statistical analyses.

## Conventional data analyses

Statistical analyses were performed using GraphPad Prism (version 8.0.1). Data normality was assessed using the Shapiro–Wilk test, which indicated non-normal distributions for all variables. Accordingly, comparisons between two independent groups were conducted using the Mann–Whitney U test, whereas comparisons involving three or more groups

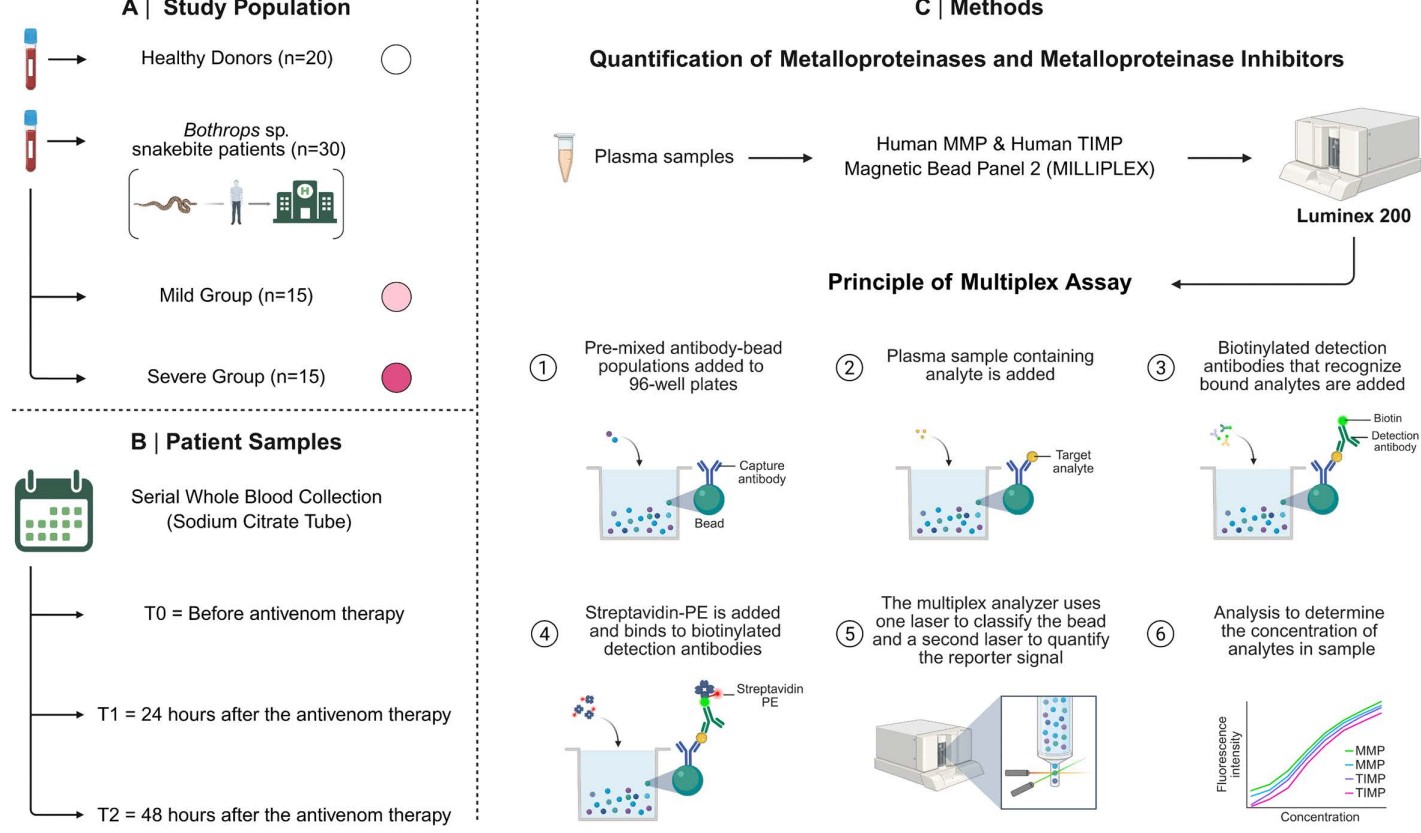

**Fig 1. Compendium of study.** The study population (A), sampling (B), and methods (C) are summarized in this figure. The study included thirty *Bothrops* snakebite patients, categorized into Mild and Severe groups based on clinical severity classification, alongside a Healthy Donor (HD) control group. Circulating levels of Matrix Metalloproteinases (MMPs) and their respective Tissue Inhibitors (TIMPs) were quantified before (T0) and post-antivenom therapy (T1 and T2). *Created in BioRender. Gama, F. (2026)* https://BioRender.com/new56x5.

employed the Kruskal–Wallis test followed by Dunn's post hoc test for multiple comparisons. Statistical significance was defined as $p < 0.05$ for all analyses.

## Curve of soluble immunological mediators

Patterns of immunological mediators and biomarker signatures were evaluated as previously described [17,18] by converting continuous analyte concentrations (pg/mL) into categorical variables. Cutoff values were defined using the global median of each analyte calculated across the complete dataset, including Healthy Donor (HD), Mild, and Severe groups. Individuals were classified as having "High" (above the cutoff) or "Low" (below the cutoff) concentrations for each molecule.

The following cutoff values were applied: MMP-1: 142.75; MMP-2: 961.16; MMP-7: 3569.90; MMP-9: 4331.14; MMP-10: 147.90; TIMP-1: 819.13; TIMP-2: 1024.84; TIMP-3: 644.17; TIMP-4: 186.00. Overall immunological signatures were visualized using radar charts, with the 50th percentile (dashed line) serving as the threshold to determine the proportion of individuals exhibiting MMP and TIMP concentrations above the global median.

The use of the global median as a cutoff was chosen to minimize the influence of outliers and non-Gaussian data distributions commonly observed in clinical biomarker datasets and to enable direct comparative evaluation of relative

expression patterns across all study groups in this exploratory analysis, rather than to establish absolute clinical thresholds.

### Integrative network of soluble immunological mediators

Biological interaction networks were constructed to explore the dynamic relationships among mediators at different follow-up time points. Integrative correlation networks were generated to characterize interactions between MMPs and TIMPs within each study group (HD, Mild, Severe). In these networks, nodes represented individual mediators, whereas edges denoted statistically significant correlations ($p < 0.05$) determined using Spearman's rank correlation. Network construction and visualization were performed using Cytoscape (version 3.10.2; Cytoscape Consortium, San Diego, CA, USA) following the developer's guidelines. Correlation strength (r) was categorized as weak ($r \leq 0.35$), moderate ($0.36 \leq r \leq 0.67$), or strong ($r \geq 0.68$), enabling detailed assessment of how MMP–TIMP relationships evolved throughout the clinical course of envenomation.

## Results

### Clinical and demographic characteristics of *Bothrops* snakebite patients stratified into mild and severe groups

Table 1 summarizes the sociodemographic and clinical characteristics of the study population. Age distribution was similar across the Mild, Severe, and HD groups. Among envenomated patients, most were male and resided in rural areas. The lower limbs were the most frequently affected anatomical site, and hypertension was the only comorbidity reported. Local manifestations clearly distinguished the clinical groups. Severe cases exhibited more extensive tissue involvement, including abscess formation and blisters (phlyctena). Notably, 54% of individuals in the Severe group developed secondary infections, reflecting a heightened susceptibility to local complications.

A strong association was observed between delayed antivenom administration and clinical severity. Patients in the Mild group received antivenom within a median of 3 hours, whereas those in the Severe group received treatment after a median of 6 hours, suggesting an association between delayed antivenom therapy and increased clinical severity of clinical outcome. Length of hospitalization further reflected disease severity. Nearly all Mild cases (93%) were discharged within 1–2 days, while all Severe cases required prolonged hospitalization, typically lasting 3–5 days.

### Profile of metalloproteinases and their inhibitors in *Bothrops* snakebite patients before antivenom administration

Analysis of metalloproteinases and their inhibitors (Fig 2) showed that the overall magnitude of the MMP/TIMP response was similar between the Mild and Severe envenomation groups prior to antivenom administration. Circulating levels of MMP-1, MMP-7, and MMP-10 were significantly elevated in both clinical groups compared with the HD controls. In contrast, MMP-9 levels were markedly reduced in envenomated patients relative to HD. Regarding endogenous inhibitors, all TIMP isoforms (TIMP-1, TIMP-2, TIMP-3, and TIMP-4) were significantly increased in both Mild and Severe groups, indicating a globally heightened regulatory response to venom-induced tissue damage. Numerical values underlying the graphical representations are provided in S1 Table.

### Dynamics of the profile of metalloproteinases and metalloproteinase inhibitors in *Bothrops* snakebite patients after the administration of antivenom

Analysis of MMP and TIMP dynamics between the Mild and Severe groups (Fig 3) revealed early differences associated with clinical severity. At baseline (T0), the Severe group exhibited a markedly dysregulated profile, with significantly higher levels of MMP-7, MMP-9, TIMP-1, TIMP-2, and TIMP-3 compared with the Mild group, indicating an exacerbated initial response to envenomation.

**Table 1. Sociodemographic and clinical characterization of the study population.**

| Variables | HD (n=20) | Mild (n=15) | Severe (n=15) | *p-value* |
|---|---|---|---|---|
| **Age** | | | | |
| Years, median (IQR) | 31 (23-63) | 39 (18-69) | 39 (18-69) | 0,806 |
| **Gender** | | | | |
| Male, n (%) | 16 (94) | 11 (73) | 14 (93) | 0,154 |
| Female, n (%) | 4 (6) | 4 (27) | 1 (7) | |
| **Zone of occurrence** | | | | |
| Urban, n (%) | – | 6 (40) | 4 (27) | 0,304 |
| Rural, n (%) | – | 9 (60) | 11 (73) | |
| **Anatomical site of snakebite, n (%)** | | | | |
| Upper, n (%) | – | 4 (27) | 2 (14) | 0,596 |
| Lower, n (%) | – | 11 (73) | 13 (86) | |
| **Comorbidities - Hypertension** | | | | |
| Yes, n (%) | – | 2 (14) – | 3 (20) | 0,570 |
| No, n (%) | | 13 (86) | 12 (80) | |
| **Local complication, n (%)** | | | | |
| Abscess, n (%) | – | – | 5 (34) | – |
| Blister, n (%) | – | – | 3 (20) | |
| **Secondary infection** | | | | |
| Yes, n (%) | – | – | 8 (54) | – |
| No, n (%) | – | – | 7 (46) | |
| **Days of hospitalization** | | | | |
| 1 - 2, n (%) | – | 14 (93) | – | 0,142 |
| 3 - 5, n (%) | – | 1 (7) | 15 (100) | |
| **Time between snakebite and administration of the antivenom** | | | | |
| 1 - 4, hours, median, IQR | – | 3 (2–4) | – | **0,035#** |
| 5 - 6, hours, median, IQR | – | – | 6 (3– 6) | |

HD, healthy donor; IQR, interquartile range. Significant differences at p<0.05 for comparisons between the mild and severe groups are represented in bold with the superscript symbol #.

Temporal patterns also differed substantially between the clinical groups. In the Mild group, MMP-1, MMP-7, and MMP-10 remained relatively stable throughout follow-up, whereas MMP-2, MMP-9, TIMP-1, TIMP-2, and TIMP-3 showed significant declines at T1 and T2 relative to T0, suggesting effective downregulation of inflammatory and ECM-remodeling pathways after antivenom administration.

In contrast, severe cases showed a persistent imbalance between MMP activity and TIMP regulation. As shown in Fig 3, MMP-2, MMP-9, TIMP-1, TIMP-2, and TIMP-3 were consistently lower at T2 compared to T0 in both Mild and Severe cases. However, MMP-1, MMP-7, and MMP-10 showed decreased concentrations only in the Severe group, sustained inhibitory response on ECM remodeling pathways and prolonged inflammatory activity.

Overall, these findings demonstrate mechanistic insight a distinct and persistently dysregulated enzymatic trajectory, characterized by heightened early activation and incomplete restoration of MMP–TIMP equilibrium. This divergence provides biological insight into how disrupted metalloproteinase dynamics may contribute to tissue destruction and worse clinical outcomes in *Bothrops* envenomation. Numerical values underlying the graphical representations are provided in S2 Table.

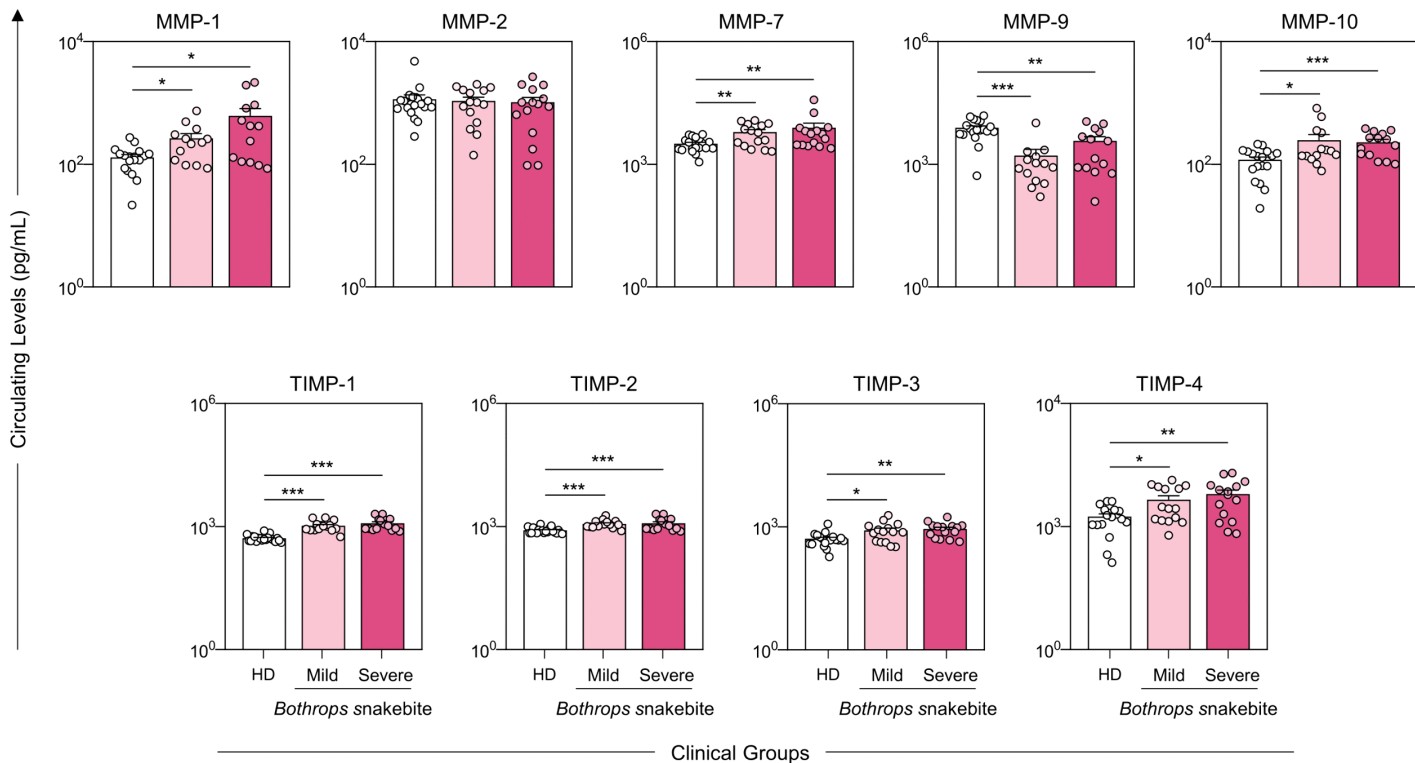

**Fig 2. Profile of metalloproteinases and metalloproteinase inhibitors in *Bothrops* snakebite patients before the administration of antivenom.**
The patients were divided into Mild (▢), Severe (▢), and HD (▢) groups. Circulating levels of Matrix Metalloproteinases (MMPs) and their respective Tissue Inhibitors (TIMPs) were quantified using a multiplex assay, as detailed in the Methods section. The results are presented using bar and symbol charts, reported in $\log^{10}$ scale, showing the median with IQR, expressed in picograms per milliliter (pg/mL). Statistical analysis was performed using the Kruskal-Wallis test followed by Dunn's post-hoc test, with significant differences denoted by asterisks: $*p < 0.05$, $**p < 0.01$, and $***p < 0.001$.

## Signature of MMPs and TIMPs in mild and severe patients

To refine the characterization of MMPs and TIMPs responses, global median values were used as cutoffs to classify individuals as low or high producers (Fig 4). This approach revealed distinct signatures across the HD, Mild, and Severe groups. The HD group showed low production of all MMPs and TIMPs, except for relatively elevated MMP-9. At T0, Mild and Severe patients exhibited high baseline levels of all TIMPs and elevated concentrations of several MMPs, indicating a strong early regulatory response

Temporal analysis showed diverging trajectories. In the Mild group, T1 and T2 follow-up demonstrated recovery of MMPs production and a progressive decline in TIMPs levels, reflecting restoration of MMPs/TIMPs balance after antivenom therapy. In contrast, Severe patients showed a consistent reduction in both MMPs and TIMPs at T1 and T2, indicating a blunted and dysregulated regulatory response. This persistent suppression contrasts with the pattern observed in Mild cases and underscores the inability of Severe patients to reestablish MMPs/TIMPs homeostasis.

## Integrative networks of *Bothrops* snakebites patients in blood samples

Integrative networks were constructed to assess interactions between MMPs and TIMPs across the HD, Mild, and Severe groups (Fig 5). In HD, the network was sparse, with only a few moderate or strong correlations such as a negative interaction between MMP-7 and TIMP-3 and a strong positive one between MMP-2 and TIMP-3—reflecting physiological ECM regulation.

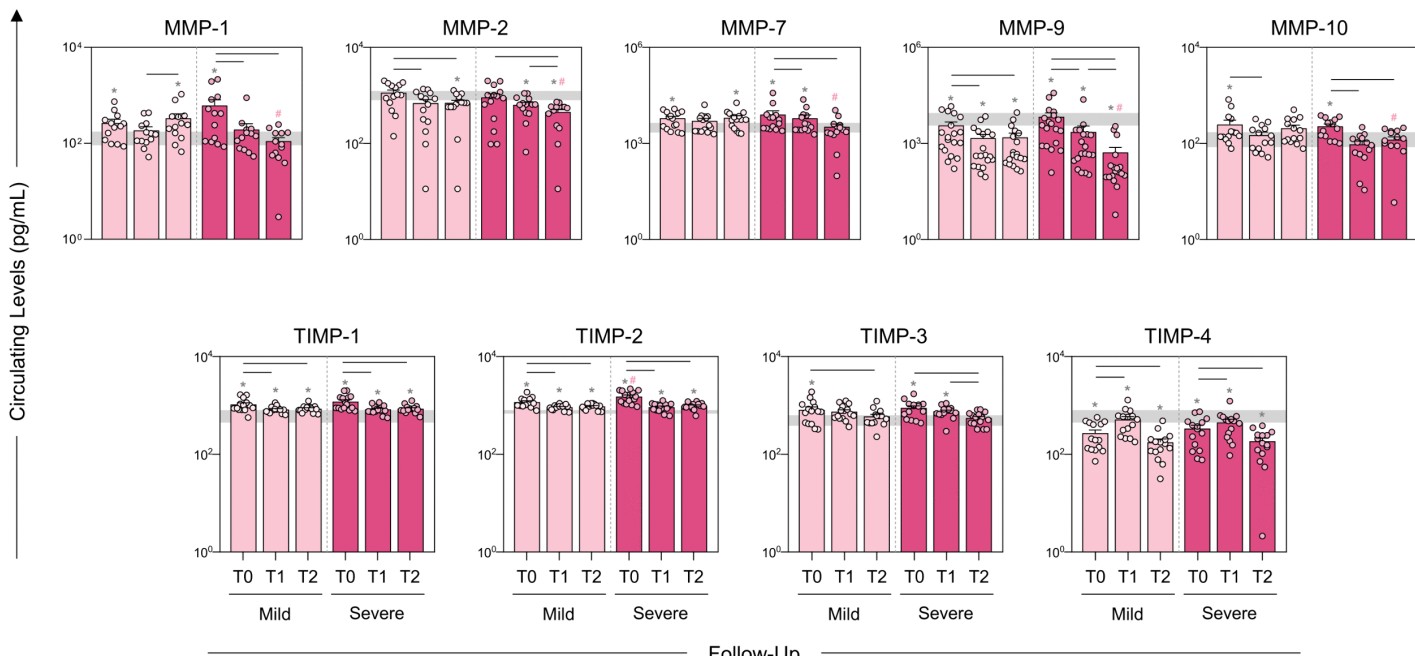

**Fig 3. Dynamics of the profile of metalloproteinases and metalloproteinase inhibitors in *Bothrops* snakebite patients after the administration of antivenom.** Circulating levels of Matrix Metalloproteinases (MMPs) and their respective Tissue Inhibitors (TIMPs) were quantified before (T0) and post-antivenom therapy (T1 and T2) in Group Mild (☐) and Severe (☐) groups. The interquartile range [25-75] of molecule concentrations in the Healthy Donor (HD) control group was used as baseline (☐). The results are presented using bar and symbol charts, reported in $\log^{10}$ scale, showing the median with IQR, expressed in picograms per milliliter (pg/mL). Statistical analyses were performed using the Wilcoxon test for comparisons between time points (T0, T1, and T2) and the Mann-Whitney U test for inter-group comparisons. Statistically significant differences ($p < 0.05$) between follow-up days are represented by the dash symbol (-). Differences compared to the HD group are highlighted with asterisks (*). Significant differences between the Mild and Severe groups are represented by the hash symbol (#).

At T0, both Mild and Severe groups showed dense and predominantly positive networks, indicating broad activation of ECM-remodeling pathways after envenomation. The Severe group displayed even stronger connectivity, with pronounced correlations among MMP-1, MMP-7, MMP-9, and TIMPs (notably TIMP-4), suggesting a more intense but poorly regulated response.

After antivenom, dynamics diverged. Mild patients preserved a highly interconnected network at both T1 and T2, consistent with inflammation resolution. In contrast, Severe patients maintained a dense network at T1 but showed clear loss of connectivity at T2. Persistent strong correlations especially within the MMP-1/MMP-7/MMP-9/MMP-10 axis and with TIMP-1 and TIMP-2 indicated ongoing dysregulation, sustained ECM degradation, and prolonged inflammation typical of severe envenomation.

## Discussion

Bothropic envenomation causes intense local and systemic effects, including edema, necrosis, hemorrhage, and coagulation disorders [1]. Metalloproteinases are central mediators of this process due to their capacity to degrade the extracellular matrix (ECM) and disrupt vascular integrity, while endogenous MMPs and TIMPs regulate ECM turnover and influence venom-induced damage [19]. An imbalance favoring MMP activity amplifies proteolysis and may worsen clinical outcomes, making these molecules potential targets for therapeutic intervention.

Clinical and epidemiological factors strongly influence prognosis, with advanced age and delayed antivenom administration consistently associated with greater severity [20]. In line with previous findings, the Severe group in our study

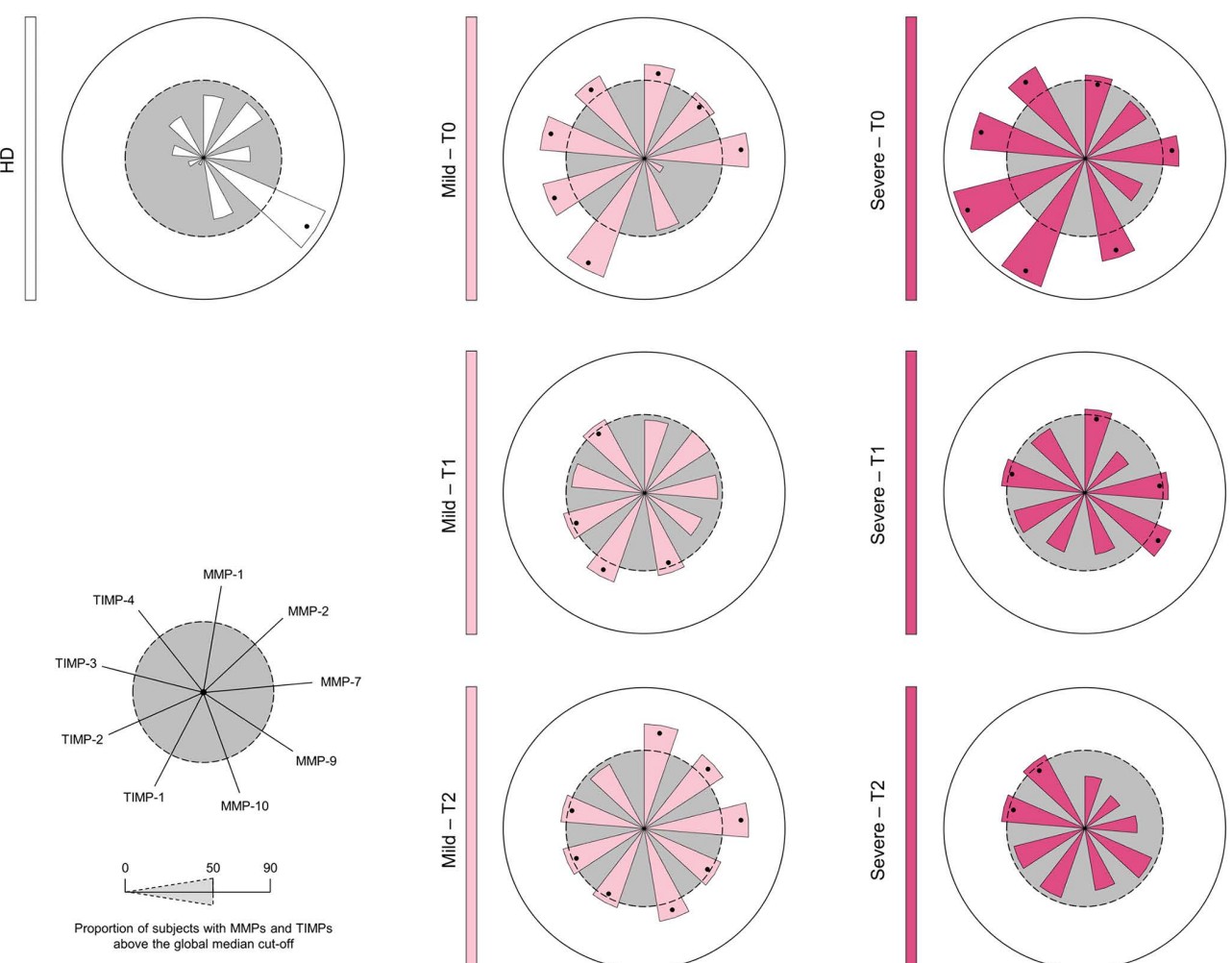

**Fig 4. Signature of metalloproteinases and metalloproteinase inhibitors in mild and severe patients.** The signature of Matrix Metalloproteinases (MMPs) and their respective Tissue Inhibitors (TIMPs) in *Bothrops* snakebite patients was constructed before (T0) and post-antivenom therapy (T1 and T2). The data, originally expressed in picograms per milliliter (pg/mL), were converted to categorical data using the overall median values as a cutoff to classify the study population as having low or high production of the MMPs and TIMPs evaluated. The overall signatures were assembled in radar charts using the 50th percentile as the threshold (central circle/gray zone) to identify MMPs and TIMPs with increased levels in a greater proportion of patients.

presented higher mean age, longer time to medical care, and a significant association between treatment delay and severity [21,22]. These variables likely shape the local progression of tissue injury and systemic inflammation, reinforcing the importance of early access to antivenom (Table 1).

*Bothrops* venom triggers a rapid and robust inflammatory response involving tissue destruction, hemorrhage, edema, and marked cytokine production [2,3]. MMP-1, MMP-7, and MMP-10 are key contributors to this process: MMP-1 degrades collagen and promotes necrosis [23,24]. MMP-7 targets basement membrane components [23], and MMP-10 amplifies inflammation and activates additional MMPs [3,25]. In our cohort, increased levels of these mediators (Fig 2) reflect active ECM degradation and inflammatory amplification. Dysregulated cytokine release, especially IL-6 and TNF-α, may further enhance MMP expression [6,7], creating a positive feedback loop that intensifies tissue injury.

The early MMP profile, characterized by increased MMP-1, MMP-7, and MMP-10 combined with reduced MMP-9, was similar in Mild and Severe patients, suggesting a uniform initial response to venom (Fig 2). Severity appears to emerge

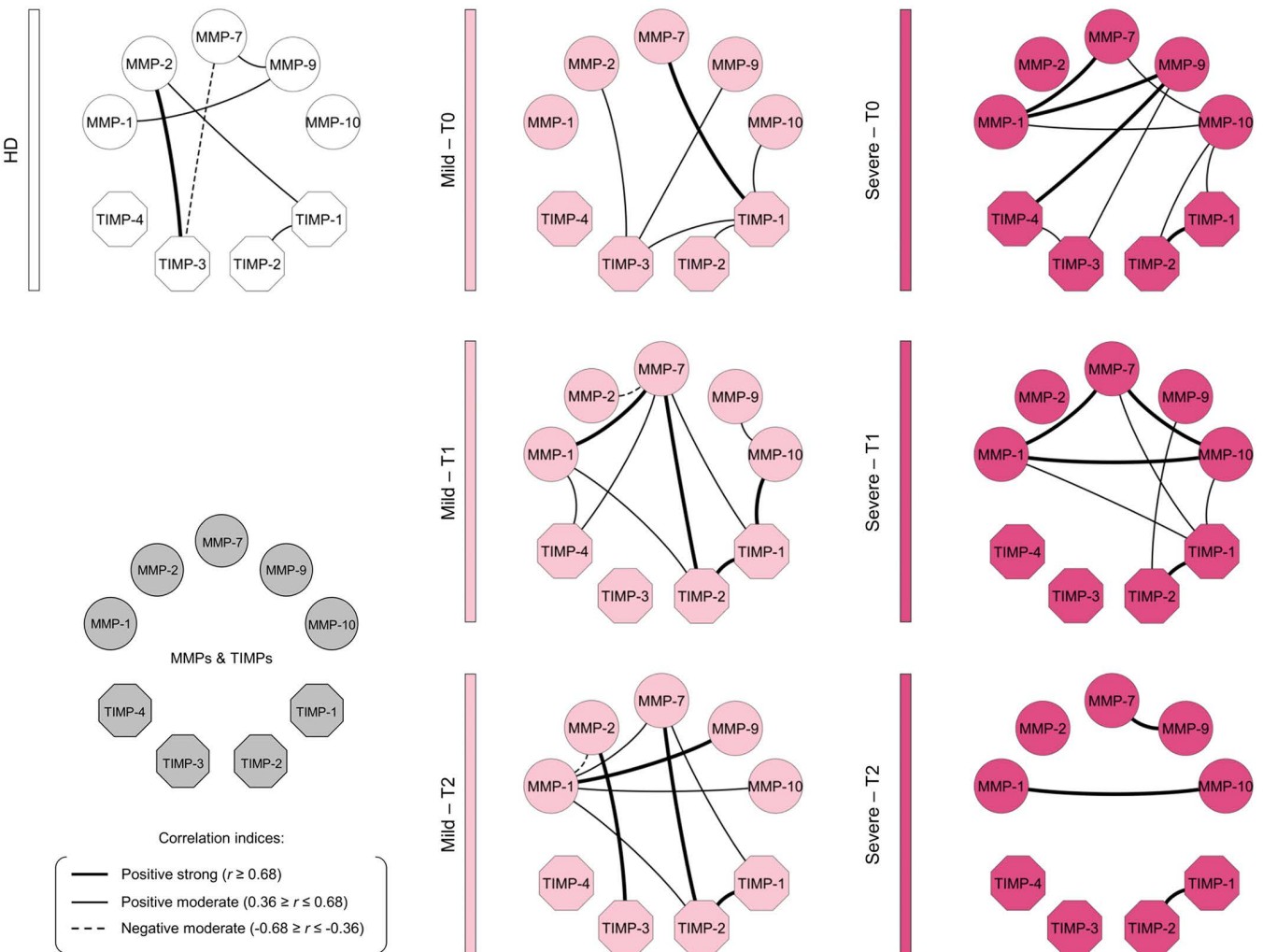

**Fig 5. Integrative networks of *Bothrops* snakebite patients in blood samples.** Networks of patients with bothropic snakebite were constructed before (T0) and post-antivenom therapy (T1 and T2) to identify the complex interactions among Matrix Metalloproteinases (MMPs) and their respective Tissue Inhibitors (TIMPs). The nodes (◯) and (◯) are used to identify the MMPs and TIMPs analyzed in the study. Solid lines between elements indicate a positive correlation, while dashed lines indicate a negative correlation. The thickness of the lines indicates the strength of the correlation. The correlation index (r) was used to categorize the strength of the correlation as weak ($r \leq 0.35$), moderate ($r \geq 0.36$ to $r \leq 0.67$), or strong ($r \geq 0.68$).

later, driven by secondary dysregulation of the MMP–TIMP axis. This reinforces the role of endogenous metalloproteinases as key determinants of tissue destruction and highlights the therapeutic potential of modulating specific MMPs or restoring MMP/TIMP equilibrium [24,26]. Such interventions may help limit the extent of local damage and reduce long-term morbidity associated with *Bothrops* envenomation.

Increased MMP activity is strongly associated with immune activation and with local and systemic injury in *Bothrops* envenomation [27], reinforcing its role in ECM degradation and disease progression [6,8,9]. However, our results show that MMP-9 levels were not elevated in either Mild or Severe groups compared with HD (Fig 2), indicating that additional mechanisms contribute to tissue damage [28]. Although TIMPs are produced to counteract excessive ECM destruction [10–12], with increased TIMP-4 observed (Fig 2), TIMP-1 and TIMP-2 responses were insufficient to contain ongoing

injury [8,13,14]. This is consistent with the fact that antivenom neutralizes circulating toxins [15], but does not reverse established local inflammation [16,17], reflected by elevated MMP-2, MMP-7, MMP-9, and MMP-10 at T0 [8,9,18,19].

The distinct behavior of MMP-1 highlights its relevance for severity. Severe patients showed a progressive reduction in MMP-1 during follow-up (Fig 3), suggesting that its initial activity may contribute to amplifying inflammation and tissue destruction [9,20]. Overall decreases in MMP levels over time likely reflect an attempt to regulate inflammation and restore homeostasis, since prolonged MMP activity is linked to excessive tissue damage and impaired healing [6,7]. Signature analysis (Fig 4) further underscored differences between groups: Mild patients displayed strong TIMPs elevation at T0, indicating an effective early regulatory response [29], whereas the Severe group's TIMPs increases were insufficient to prevent clinical progression, pointing to early disruption of the MMPs–TIMPs axis.

Post-antivenom dynamics further distinguished recovery patterns. In Mild patients, increases in MMP-2, MMP-7, and MMP-10 after treatment suggest activation of ECM repair and remodeling once venom components are neutralized [23]. In contrast, Severe patients showed minimal post-treatment MMPs activation, suggesting an inability to mount an effective repair response, potentially due to extensive early damage or depletion of key regenerative mediators [30,31]. This divergence supports the hypothesis that a higher venom load or individual susceptibility may overwhelm repair pathways, preventing adequate mobilization of MMP-dependent healing processes [21–24].

Our results (Fig 5) show that under basal conditions (HD), immune interactions are tightly regulated, as evidenced by the negative MMP-2/TIMP-3 correlation, which reflects homeostatic ECM control through targeted inhibition of gelatinase activity [32,33]. In contrast, *Bothrops* envenomation induces an immediate restructuring of the correlation network at T0 in both patient groups, indicating systemic activation in response to venom components and early ECM degradation [29]. Strong positive correlations among MMP-7, MMP-9, and TIMP-1, TIMP-2, and TIMP-3 reflect an acute inflammatory response characterized by simultaneous upregulation of proteases and inhibitors [34,35]. The altered and inefficient network structure observed in the Severe group suggests an ineffective regulatory response unable to contain ongoing tissue injury, contributing to unfavorable clinical evolution [36].

After antivenom administration, the evolution of the MMPs/TIMPs network becomes a key indicator of clinical resolution [37]. In the Mild group, the marked reduction in network complexity at T2 reflects effective inflammatory control and restoration of homeostasis, consistent with transition to a reparative phase [31]. Conversely, the Severe group maintains a weak and poorly integrated network at T1 and T2, with persistent interactions among MMP-1, MMP-7, MMP-9, MMP-10, and their inhibitors (Fig 5). This sustained disorganization suggests a prolonged imbalance in the tissue microenvironment, where ongoing protease activation and insufficient inhibition perpetuate pathology [30,36,38].

Overall, the findings of this study suggest that the ability to modulate and simplify MMP/TIMP interaction patterns over time may be associated with more favorable clinical evolution in *Bothrops* envenomation. In this context, Mild cases tended to exhibit a progressive attenuation of network complexity following antivenom administration, whereas Severe cases were characterized by more persistent and heterogeneous interaction patterns.

Based on these observations, we propose a conceptual model summarizing the temporal dynamics of MMPs and TIMPs during envenomation (Fig 6). At T0, venom-induced tissue injury is accompanied by an early inflammatory response and activation of extracellular matrix remodeling pathways. By T1, 24 hours after antivenom administration, differences in clinical severity become apparent and appear to coincide with distinct patterns of MMP/TIMP regulation. Finally, at T2, the degree to which regulatory balance is restored or remains disrupted may reflect divergent trajectories of tissue repair. This model is intended to be descriptive and hypothesis-generating rather than mechanistic.

This study has important limitations that should be considered when interpreting the findings. The relatively small sample size reflects the inherent challenges of conducting prospective clinical studies in real-world snakebite settings and limits the ability to perform extensive stratified or multivariable analyses. In addition, the integrative network analyses and correlation-based visualizations employed here are descriptive in nature and do not imply causal relationships between individual mediators.

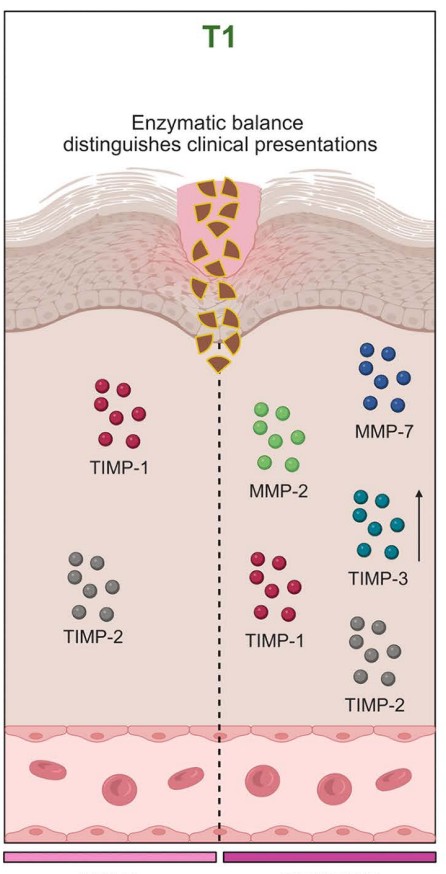
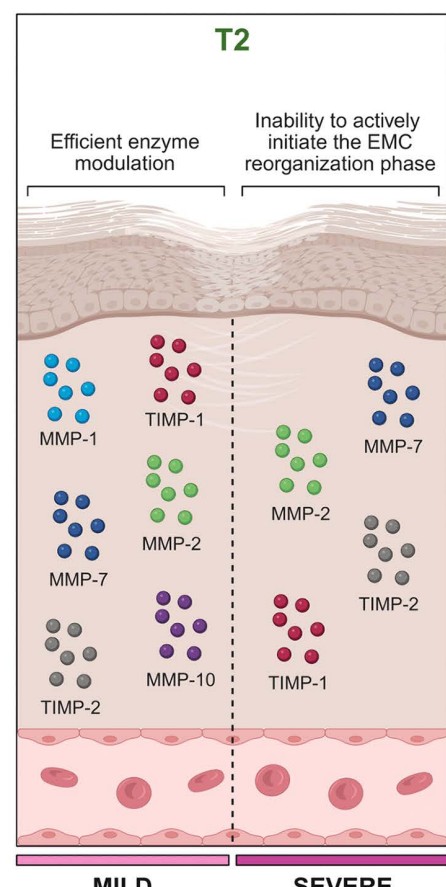

**① Initial activation of the inflammatory response**

**② Differentiation of severity based on the imbalance of regulators**

**③ Resolution, modulation and homeostasis**

**Fig 6. Schematic summary of the local inflammatory response to *Bothrops* envenomation in Mild and Severe patients.** This model illustrates the Matrix Metalloproteinases (MMPs) and their respective Tissue Inhibitors (TIMPs) dynamics in *Bothrops* envenomation, from inoculation to tissue repair. (A) Inflammatory Response and Initial Activation (T0): Venom inoculation induces acute damage and the release of MMPs (MMP-1, -2, -7, -10) and TIMPs, establishing an initial proteolytic imbalance. (B) Severity Differentiation (T1): 24 hours after antivenom therapy, clinical progression is distinguished by enzymatic modulation. The Mild group exhibits efficient regulatory balance (TIMPs control MMPs), whereas the Severe group maintains a persistent imbalance (predominance of MMPs, such as MMP-9). (C) Resolution and Homeostasis (T2): In the Mild group, modulation is sufficient, with MMP-10 (an activator) and TIMPs promoting matrix extracellular (ECM) turnover and repair. The Severe group fails to reorganize the ECM, indicating dysfunctional and prolonged recovery. *Created in BioRender. Gama,* F. *(2026)* https://BioRender.com/ly48v07.

Furthermore, this investigation focused exclusively on circulating MMPs and their inhibitors and did not assess other inflammatory mediators, such as cytokines, chemokines, complement components, or venom-derived proteases, which are also known to influence tissue injury and repair processes. Accordingly, the associations observed here should be interpreted as exploratory and hypothesis-generating. Future studies incorporating larger cohorts and broader immunological profiling will be essential to validate and extend the hypotheses proposed by the present findings.

## Conclusion

Characterizing the circulating profiles of metalloproteinases (MMP-1, -2, -7, -9, -10) and their inhibitors (TIMP-1, -2, -3, -4) in patients with suspected Bothrops envenomation provides insight into the dynamics of inflammation and its association

with clinical severity. Our findings indicate that although early alterations in MMP and TIMP levels at T0 were broadly observed across patients, subsequent temporal patterns differed between Mild and Severe cases. Mild cases tended to exhibit more effective modulation of MMP/TIMP balance over time, whereas Severe cases showed more persistent dysregulation, which may be associated with impaired tissue remodeling.

Overall, this study provides exploratory evidence that circulating MMP and TIMP profiles undergo dynamic modulation during the clinical course of snakebite envenomation. Although the observed variations were relatively subtle, the longitudinal and integrative analyses suggest distinct regulatory trajectories associated with clinical severity. These findings should be interpreted as hypothesis-generating and highlight the need for future studies incorporating larger cohorts and additional inflammatory and venom-related parameters to more comprehensively define the mechanisms underlying tissue injury and repair following snakebite.

## Supporting information

**S1 Table. Circulating levels of MMPs and TIMPs in healthy donors and patients with mild or severe suspected Bothrops envenomation at baseline (T0).**
(DOCX)

**S2 Table. Longitudinal circulating levels of MMPs and TIMPs in healthy donors and in patients with mild or severe suspected Bothrops envenomation at baseline (T0) and after antivenom therapy (T1 and T2).**
(DOCX)

## Acknowledgments

We would like to thank all the authors and researchers at FMT-HVD, HEMOAM, UEA, and UFAM for their critical discussions and insightful and encouraging ideas. We are also grateful for the support and thoughts that helped shape our intuition and the perspectives that are highlighted in this manuscript. The authors thank the Program for Technological Development in Tools for Health – RPT-FIOCRUZ for use of the Flow Cytometry facility at FIOCRUZ-Minas (RPT08D) and the financial support provided by FAPEMIG (APQ-03113–24).

## Author contributions

**Conceptualization:** Juliana Costa Ferreira Neves, Jacqueline Almeida Gonçalves Sachett, Olindo Assis Martins-Filho, Andréa Teixeira-Carvalho, Marco Aurélio Sartim, Wuelton Monteiro, Allyson Guimarães Costa.

**Data curation:** Juliana Costa Ferreira Neves, Fábio Magalhães-Gama, Hiochelson Najibe Santos Ibiapina, Êndila Souza Barbosa.

**Formal analysis:** Juliana Costa Ferreira Neves, Fábio Magalhães-Gama, Ana Carolina Campi-Azevedo, Andréa Teixeira-Carvalho.

**Funding acquisition:** Olindo Assis Martins-Filho, Wuelton Monteiro, Allyson Guimarães Costa.

**Investigation:** Juliana Costa Ferreira Neves, Hiochelson Najibe Santos Ibiapina, Kamille Beltrão Seixas, Êndila Souza Barbosa.

**Methodology:** Juliana Costa Ferreira Neves, Fábio Magalhães-Gama, Ana Carolina Campi-Azevedo, Olindo Assis Martins-Filho, Andréa Teixeira-Carvalho, Allyson Guimarães Costa.

**Project administration:** Allyson Guimarães Costa.

**Resources:** Kamille Beltrão Seixas, Adriana Malheiro, Jacqueline Almeida Gonçalves Sachett, Wuelton Monteiro.

**Supervision:** Adriana Malheiro, Jacqueline Almeida Gonçalves Sachett, Olindo Assis Martins-Filho, Andréa Teixeira-Carvalho, Marco Aurélio Sartim, Wuelton Monteiro, Allyson Guimarães Costa.

**Visualization:** Fábio Magalhães-Gama, Ana Carolina Campi-Azevedo.

**Writing – original draft:** Juliana Costa Ferreira Neves, Olindo Assis Martins-Filho, Andréa Teixeira-Carvalho, Allyson Guimarães Costa.

**Writing – review & editing:** Juliana Costa Ferreira Neves, Fábio Magalhães-Gama, Hiochelson Najibe Santos Ibiapina, Kamille Beltrão Seixas, Êndila Souza Barbosa, Adriana Malheiro, Jacqueline Almeida Gonçalves Sachett, Ana Carolina Campi-Azevedo, Olindo Assis Martins-Filho, Andréa Teixeira-Carvalho, Marco Aurélio Sartim, Wuelton Monteiro, Allyson Guimarães Costa.

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
