## [Decision Letter · Decision Letter 0]

1 Feb 2026

Veja o que você faz meus tecidos fazerem: o papel das metaloproteinases e seus inibidores em mordidas de cobra do gênero Bothrops

Dear Dr. Costa,

Thank you for submitting your manuscript to PLOS Neglected Tropical Diseases. After careful consideration, we feel that it has merit but does not fully meet PLOS Neglected Tropical Diseases's publication criteria as it currently stands. Therefore, we invite you to submit a revised version of the manuscript that addresses the points raised during the review process.

Please submit your revised manuscript within by Apr 02 2026 11:59PM. If you will need more time than this to complete your revisions, please reply to this message or contact the journal office at plosntds@plos.org. Please include the following items when submitting your revised manuscript:

We look forward to receiving your revised manuscript.

Kind regards,

Hatem Kallel

Academic Editor

José María Gutiérrez

Section Editor

Shaden Kamhawi

co-Editor-in-Chief

Paul Brindley

co-Editor-in-Chief

**Journal Requirments**:

At this stage, the following Authors/Authors require contributions: Juliana Costa Ferreira Neves, Fábio Magalhães-Gama, Hiochelson Najibe Santos Ibiapina, Kamille Beltrão Seixas, Êndila Souza Barbosa, Adriana Malheiro, Jacqueline de Almeida Gonçalves Sachett, Ana Carolina Campi-Azevedo, Olindo Assis Martins-Filho, Andréa Teixeira-Carvalho, Marco Aurélio Sartim, Wuelton Monteiro, and Allyson Guimarães Costa. Please ensure that the full contributions of each author are acknowledged in the "Add/Edit/Remove Authors" section of our submission form.

3) Please ensure that the Title in your manuscript file and the Title provided in your online submission form are the same.

4) We do not publish any copyright or trademark symbols that usually accompany proprietary names, eg ©,  ®, or TM  (e.g. next to drug or reagent names). Therefore please remove all instances of trademark/copyright symbols throughout the text, including:

- ® on page: 4.

Potential Copyright Issues:

i) Figures 1, and 6. Please confirm whether you drew the images / clip-art within the figure panels by hand. If you did not draw the images, please provide (a) a link to the source of the images or icons and their license / terms of use; or (b) written permission from the copyright holder to publish the images or icons under our CC BY 4.0 license. Alternatively, you may replace the images with open source alternatives. See these open source resources you may use to replace images / clip-art:

- https://openclipart.org/....

6) Please provide a complete Data Availability Statement in the submission form, ensuring you include all necessary access information or a reason for why you are unable to make your data freely accessible. If your research concerns only data provided within your submission, please write "All data are in the manuscript and/or supporting information files" as your Data Availability Statement.

7) Please send a completed 'Competing Interests' statement, including any COIs declared by your co-authors. If you have no competing interests to declare, please state "The authors have declared that no competing interests exist". Otherwise please declare all competing interests beginning with the statement "I have read the journal's policy and the authors of this manuscript have the following competing interests"

**Reviewers' Comments:**

Reviewer's Responses to Questions

**Key Review Criteria Required for Acceptance?**

**Methods**

-Are the objectives of the study clearly articulated with a clear testable hypothesis stated?

-Is the study design appropriate to address the stated objectives?

-Is the population clearly described and appropriate for the hypothesis being tested?

-Is the sample size sufficient to ensure adequate power to address the hypothesis being tested?

-Were correct statistical analysis used to support conclusions?

-Are there concerns about ethical or regulatory requirements being met?

Reviewer #1: 2.3 – duplication (with minor differences) of the first two paragraphs. How was Bothrops envenoming identified? This isn’t described, and if no formal diagnosis, edit to “suspected Bothrops sp.”

2.3 - It would be helpful to contextualise the T0 timepoint. Is this shortly after admission? How much variation was there in terms of time from bite to T0 – please provide a range.

2.4 – This section needs information on replicates, both in terms of technical and independent replicates, where applicable.

2.6 – What was the rationale for using median of the three groups as the cutoff value rather than the mean (or mean + a factor of the standard deviation) of the HD control?

Reviewer #2: (No Response)

**Results**

-Does the analysis presented match the analysis plan?

-Are the results clearly and completely presented?

-Are the figures (Tables, Images) of sufficient quality for clarity?

Reviewer #1: Figure 3 – significant differences needs to be better shown. This might be currently hampered by the figure resolution.

Reviewer #2: The graphics are mostly low-resolution, making it impossible to read the values on the Y axis.

I suggest having a supplementary table containing the raw values per experiment.

**Conclusions**

-Are the conclusions supported by the data presented?

-Are the limitations of analysis clearly described?

-Do the authors discuss how these data can be helpful to advance our understanding of the topic under study?

-Is public health relevance addressed?

Reviewer #1: More discussion of the limitations are required.

Much of the discussion contains assertions based on predictions of the visualised correlations, and I think the authors would be better placed to reword these sections to highlight that this study provides valuable pilot data that will enable them (or others) to more robustly test explicit hypotheses in future studies with larger sample sizes.

Reviewer #2: I consider that the conclusions are just partially supported by the data presented. As explained in the general comments, the variation in MMP/TIMP appears subtle and should be complemented by investigating additional parameters in the individuals.

**Editorial and Data Presentation Modifications?**

Reviewer #1: (No Response)

Reviewer #2: Note, that on lines 119-129 there are duplicated information on both paragraphs

**Summary and General Comments**

Reviewer #1: Overall this is an interesting study, which is pretty novel for the field of snakebite. The authors quantify key MMPs and TIMPs from clinical samples collected from envenomed patients in the Brazilian Amazon and interrogate the resulting dataset in the context of mild and severe stratified patients. Further elaboration of the methods is required, and I think a more robust discussion section that draws more widely on the literature, further highlights limitations, and perhaps explains what key next steps are required to validate the hypotheses generated from this data, would make the paper more impactful to the field.

Reviewer #2: The article addresses an important aspect of snake envenomation: the secondary activation of endogenous proteinases, in this case MMPs, which may exacerbate its effects or mediate recovery. The experimental approach was based on measuring MMP and TIMPs (MMP-inhibitors) in healthy or envenomated human subjects (treated or not with the antiserum) using standard methodologies. No other mediators or specific pathologic effects were evaluated.

The results indicate a discrete alteration in MMP and TIMP levels upon envenoming, slightly associated with the level of severity. Most of the comparisons between cohorts were shown to be statistically significant according to the tests applied. However, the values indicated in the graphs seem to be generally very similar among the groups and variables within samples. Since the study is based on various clinical samples, especially from real bite accidents that may vary in effectiveness, amount of venom injected, etc., it is expected to obtain such heterogeneous values per group. However, with such minor variation in MMP and TIMP, and the absence of information about other mediators, it is very hard to interpret the results and draw the inflammation model proposed.

I suggest presenting the measured values in a separate (supplementary) table and demonstrating the statistical tests performed. And, to better sustain the conclusions about the mechanisms involved uppon MMP activation, other inflammatory indicators should be evaluated.

Nevertheless, the article provides strong evidence for the activation of MMPs and the timeline of the secondary steps of envenomation. This is an important contribution to the dynamics of snake envenoming and to the development of complementary treatments.

PLOS authors have the option to publish the peer review history of their article (what does this mean?). If published, this will include your full peer review and any attached files.). If published, this will include your full peer review and any attached files.). If published, this will include your full peer review and any attached files.). If published, this will include your full peer review and any attached files.

...

Reviewer #1: No

Reviewer #2: No

**Figure resubmission:**
---

## [Editor Report · Decision Letter 1]

9 Apr 2026

Look What You Make My Tissues Do: The Role of Metalloproteinases and Their Inhibitors in Bothrops Snakebites

Dear Dr. Costa,

Thank you for submitting your manuscript to PLOS Neglected Tropical Diseases. After careful consideration, we feel that it has merit but does not fully meet PLOS Neglected Tropical Diseases's publication criteria as it currently stands. Therefore, we invite you to submit a revised version of the manuscript that addresses the points raised during the review process.

Please submit your revised manuscript within by May 09 2026 11:59PM. If you will need more time than this to complete your revisions, please reply to this message or contact the journal office at plosntds@plos.org. Please include the following items when submitting your revised manuscript:

We look forward to receiving your revised manuscript.

Kind regards,

José María Gutiérrez

Section Editor

Shaden Kamhawi

co-Editor-in-Chief

Paul Brindley

co-Editor-in-Chief

**Editor Comments:**

The reviewers of this manuscript valued the novelty of the study and its relevance to understand the complex dynamics of the inflammatory response to snakebite envenoming in samples from patients. At the same time, they highlighted a number of issues that should be considered for preparing a revised version, including a more detailed description of methods, a refinement of the discussion, with further details about the limitations of the study, and the inclusion of the raw data in a supplementary table. Also, the authors may consider the inclusion of additional inflammatory parameters to make the analysis stronger.

**Journal Requirements:**

1) Please ensure that the funders and grant numbers match between the Financial Disclosure field and the Funding Information tab in your submission form. Note that the funders must be provided in the same order in both places as well.

**Figure resubmission:**
---

## [Editor Report · Decision Letter 2]

14 Apr 2026

Dear Dr Costa,

We are pleased to inform you that your manuscript 'Look What You Make My Tissues Do: The Role of Metalloproteinases and Their Inhibitors in Bothrops Snakebites' has been provisionally accepted for publication in PLOS Neglected Tropical Diseases.

Best regards,

José María Gutiérrez

Section Editor

Shaden Kamhawi

co-Editor-in-Chief

Paul Brindley

co-Editor-in-Chief

---

## [Editor Report · Acceptance letter]

Dear Dr. Costa,

We are delighted to inform you that your manuscript, "Look What You Make My Tissues Do: The Role of Metalloproteinases and Their Inhibitors in Bothrops Snakebites," has been formally accepted for publication in PLOS Neglected Tropical Diseases.

Best regards,

Shaden Kamhawi

co-Editor-in-Chief

Paul Brindley

co-Editor-in-Chief
